

# Sonification of reference markers for auditory graphs: effects on non-visual point estimation tasks

Oussama Metatla, Nick Bryan-Kinns, Tony Stockman and Fiore Martin

School of Electronic Engineering and Computer Science, Queen Mary University of London, United Kingdom

## ABSTRACT

Research has suggested that adding contextual information such as reference markers to data sonification can improve interaction with auditory graphs. This paper presents results of an experiment that contributes to quantifying and analysing the extent of such benefits for an integral part of interacting with graphed data: point estimation tasks. We examine three pitch-based sonification mappings; pitch-only, one-reference, and multiple-references that we designed to provide information about distance from an origin. We assess the effects of these sonifications on users' performances when completing point estimation tasks in a between-subject experimental design against visual and speech control conditions. Results showed that the addition of reference tones increases users accuracy with a trade-off for task completion times, and that the multiple-references mapping is particularly effective when dealing with points that are positioned at the midrange of a given axis.

## INTRODUCTION

Graphs are a prevalent means for presenting information. Their advantages over other forms of representation, such as text, for visual data display and analysis have been thoroughly demonstrated (e.g., *Larkin & Simon, 1987*; *Tufte & Graves-Morris, 1983*). The ability to construct graphs is also critical in helping users not only visualise complex concepts, but also promote learning by doing (*Bransford, Brown & Cocking, 1999*). However, there are many situations where visual displays can be inadequate for accessing information. For example, when users engage in multiple tasks that compete for visual attention, or for individuals who experience a situational or permanent visual impairment (*Kramer, 1994*). Sonification has been the focus of increasing research as a means for providing non-visual access to data, including graphs, by displaying data using non-speech sounds. In particular, developments in the field have found that the auditory system is well suited to detect patterns in data sets similar to those represented by visual graphs, allowing listeners to perceive and actively engage with data structures and properties (*Kramer, 1994*; *Flowers & Hauer, 2005*). In this context, we are interested in exploring support for non-visual point estimation tasks since they form an integral part of editing and interpreting graphed

Corresponding author
Oussama Metatla,
o.metatla@qmul.ac.uk

data (*Tufte & Graves-Morris, 1983*). In particular, we are interested in exploring how adding contextual information to data sonification can improve support for such tasks. Previous research has suggested that adding contextual information such as reference markers can improve interaction with sonified graphs (e.g., *Smith & Walker, 2005*). This paper presents an experiment that contributes to quantifying and analysing the extent of such benefits for non-visual point estimation tasks by contrasting three pitch-based sonification mappings; pitch-only, one-reference, and multiple-references. The contrasting of these sonification techniques is explored in comparison to visual and speech-based displays. The reported results have implications for ongoing research into alternative information displays in general, and auditory displays in particular. Investigating factors affecting performance will lead to better design of auditory graphs and other alternative displays in ways that enhance user performance and flexibility, and allow the improvement of system interfaces wherever such displays are required.

## BACKGROUND

### Auditory graphs

In their basic form, auditory graphs are produced by mapping data values on the visual $X$ and $Y$ axes to auditory dimensions, such as frequency and amplitude (*Brown et al., 2003*). This basic sonification technique has been successfully used to provide non-visual access to a variety of visual graph-based representations including line graphs (*Mansur, Blattner & Joy, 1985*), seismograms (*Hayward, 1992*) and time series data (*Flowers & Hauer, 1995*). *Mansur, Blattner & Joy (1985)* pioneered the technique when they developed *sound graphs*, which mapped data values on the $y$-axis of a line graph to continuous pitch and the $x$-axis to temporal presentation. They found that, after a small amount of training, users were able to identify key patterns in the underlying data such as linearity and symmetry on 79%–95% of the trials. Similarly, *Flowers & Hauer, (1995)* conducted a series of studies in which they examined non-visual presentation of statistical data by combining various dimensions of sound. For instance, they used pitch to represent the $y$-axis of a polygon and loudness for the values on the $x$-axis and found that auditory scatter plots are as efficient as visual representations in conveying the sign and magnitude of correlations.

The issue of how to map the dimensions of sound to the data being represented is at the core of auditory graph design. For instance, whether to increase or decrease a perceptual dimension such as pitch in response to changes in the underlying data. *Brown et al. (2003)* examined such issues and produced guidelines for auditory graph design grounded in research into the sonification of line graphs, in addition to guidelines specific to the sonification of graphs containing two or three data series. *Walker & Mauney (2010)* explored preferred data-to-display mappings, polarities, and scaling functions to relate data values to underlying sound parameters for both sighted and visually impaired listeners and found general agreement about polarities obtained with the two listener populations in most studied cases. In terms of compatibility with other presentation modalities, *Nees & Walker (2007)* argued that pitch mappings allow for the emergence of patterns in data and showed that perceptual grouping of tones could act much like the primary display

advantage of visual graphs, which lies in their ability to efficiently communicate unnoticed patterns. Early studies of auditory graphs have also found them to be comparable in efficacy to tactile displays (*Mansur, Blattner & Joy, 1985*), with tactile displays yielding slightly more accurate responses and auditory graphs resulting in faster reaction times. *Bonebright et al. (2001)* determined that, in general, users are able to match an auditory graph to a visual line graph or scatter plot of the same data. *Brown et al. (2003)* also found that people could produce a visual rendition of a graph that was over 80% accurate (on average) after hearing an auditory presentation, and *Harrar & Stockman (2007)* found that a continuous display of auditory graphs produced more accurate visual renderings when compared against the use of discrete tones.

## Point estimation in auditory graphs

However, researchers soon realised that there is more to designing effective auditory graphs than merely dealing with the issues of data-to-sound mappings. Whilst presenting quantitative data, visual graphs also present a rich set of information that helps improve the readability and comprehension of such data. In visual information display, additional information such as axes, labels and tick marks increases readability and aids perception by enabling more effective top-down processing (*Smith & Walker, 2002*). A visual graph without context cues (e.g., no axes) provides no way to estimate values at any point. It is these kinds of characteristics that give visual graphs advantages over other means of information presentation, such as linear textual forms (*Larkin & Simon, 1987*).

A common method for adding $x$-axis context to a sonification is to use a series of clicks or percussive sounds. *Bonebright et al. (2001)* investigated the use of rhythmic markers in the form of click sounds and explored whether students could match auditory representations with the correct visual graphs. Graph reading tasks, such as point estimation, which form the focus of this paper, can be greatly effected by the lack of context and reference information. For instance, *Nees & Walker (2008)* examined the role of data density (i.e., the number of discrete data points presented per second) and trend reversals for both point-estimation and trend-identification tasks with auditory graphs. For the point estimation tasks, they found that users' performance declined with increased data density and trend reversals. *Smith & Walker (2005)* investigated how adding a variety of contextual information can improve non-visual point estimation tasks in such cases. They explored the use of click sounds to represent context on the $x$-axis and the addition of reference markers that provide scaling cues on the $y$-axis and found that the addition of auditory context enhances the interpretation of auditory graphs. This line of research has shown that representing data through auditory graphs will be more effective if context information is included and properly designed. Further studies are needed to investigate possible methods for implementing context in order to allow users of sonifications to go beyond the tasks of trend analysis and also to be able to perform point estimation tasks effectively. Moreover, previous studies have mostly focused on passive listening. For example, to explore financial data, users listened to whole graph sonifications before estimating values at certain points of interest (*Smith & Walker, 2002*). The experiment presented in this paper explores whether

such benefits extend to interactive sonification, i.e., where users have active control over the audio output as they explore data on an axis.

## Pointing in haptic and tactile interaction

Other research has focused on haptic and tactile displays as a means for target acquisition at the user interface, although not specifically for point estimation tasks. Pointing, as a gesture for indicating direction or focus, and proprioception are indeed a natural fit for exploring haptic interaction at the user interface. Using a Fitt's model (*Fitts, 1954*), *Ahmaniemi & Lantz (2009)* explored the use of tactile feedback to support pointing in augmented reality applications and found that the width and distance of a target had a significant effect on pointing and probing times. However, their findings also showed how Fitt's law is not an adequate model for point estimation because it does not account for the strategy employed by users when searching for targets. Focusing on accessibility, *Li, Dearman & Truong (2010)* explored the use of proprioception to support visually impaired users to rearrange icons in a virtual space through a mobile interactive prototype that leverage accelerometer and gyroscope data. *Fiannaca, Morelli & Folmer (2013)* also used a mobile device to explore how proprioception coupled with haptic feedback can support interaction with invisible objects. Their technique uses haptic feedback to position the user's arm and hand to point out the location of a virtual object. *Gustafson, Bierwirth & Baudisch (2010)* investigated imaginary interfaces, which are screen-less devices that allow users to perform spatial interaction with empty hands and without visual feedback, combining both touch and gestural interaction. Their evaluations of using this technique for drawing and pointing to locations showed that users' visual short-term memory can, in part, replace the feedback conventionally displayed on a screen.

## Non-visual graph editing

Although research on non-visual access to graphs is steadily growing, relatively little work has investigated strategies for actively constructing and editing such graphs through non-visual means. There are of course manual solutions for doing so, using physical artefacts such as pins and cardboard (*Brookshire, 2006*), but these can be inadequate for handling complex graphics and do not allow for flexible storage, editing and reproduction. *McGookin, Robertson & Brewster (2010)* examined how some of these issues could be addressed through tangible user interface design and developed the Tangible Graph Builder to allow for constructing and browsing of chart-based data. Most computer-based solutions to non-visual graph editing combine audio and haptic technologies. For instance, *McGookin & Brewster (2007)* developed an audio-haptic application for constructing bar graphs and *Bernareggi et al. (2008)* developed an interactive system to create, edit and explore graph structures through direct manipulation operations using audio-haptic interaction, supported by visual feedback. More recently, *Metatla et al. (2012b)* developed a cross-modal diagram editor to support collaboration between visually-impaired and sighted coworkers using virtual haptic and non-speech audio techniques. They also explored how connected graphs can be edited using audio-only interaction (*Metatla, Bryan-Kinns & Stockman, 2012*). Evaluations of these applications show that users could effectively construct and

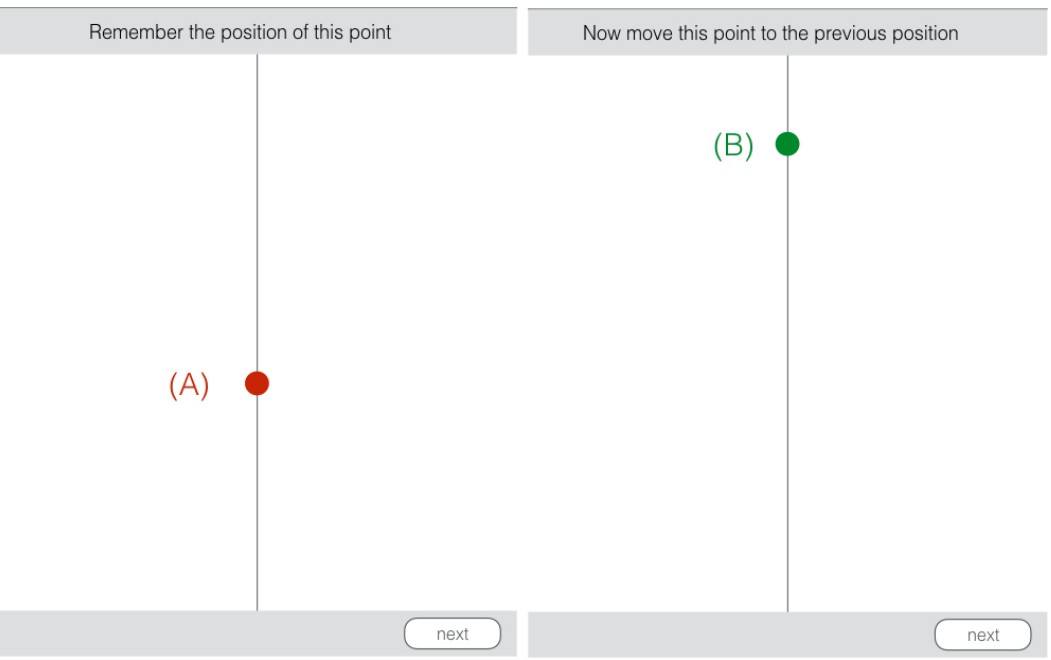

**Figure 1** **Experimental task.** Point estimation task used in the experiment; participants were first asked to remember the location of a target position (A) and then to reposition a second point (B) from a randomly generated starting point on the $y$-axis back to the previously memorised target position.

manipulate graphical representations non-visually. There is, however, little research that looks at supporting interactive editing of auditory graphs based on sonification or that looks at supporting point estimation tasks through active as opposed to passive listening. This paper contributes to addressing this gap in the literature by exploring the impact of different types of sonifications of reference markers on point estimation tasks.

## EXPERIMENT

Previous studies have shown that the addition of context cues, such as tick marks and labels can have added benefits for non-visual interaction with graphs (e.g., *Smith & Walker, 2002*; *Nees & Walker, 2008*). Estimating the position of a point in space forms an integral part of reading and/or constructing graph-based representations (c.f., *Larkin & Simon, 1987*; *Tufte & Graves-Morris, 1983*). We therefore designed an experiment to investigate what effect using sonification to add contextual reference markers has on the accuracy of estimating point positions. We focused our investigation on conveying information that could help when estimating the position of a point in terms of its distance from an origin point.

### Apparatus

We designed a simple user interface to support the task of editing the position of a point, focusing on the part where users need to estimate the position of a point when placing it at a desired location on an axis (see Fig. 1). The interface allows users to manipulate the position of a point using the keyboard up and down arrow keys on an axis containing a total of 30 positions (ranging from −15 to 15, the value 0 being the middle position).

**Table 1 Sonification.** Pitch values (in Hz) used to sonify the 30 points scale.

| Position | Pitch | Position | Pitch | Position | Pitch | Position | Pitch |
|---|---|---|---|---|---|---|---|
| −15 | 120 | −7 | 324 | 1 | 877 | 9 | 2,371 |
| −14 | 135 | −6 | 367 | 2 | 993 | 10 | 2,685 |
| −13 | 153 | −5 | 416 | 3 | 1,124 | 11 | 3,040 |
| −12 | 174 | −4 | 471 | 4 | 1,273 | 12 | 3,443 |
| −11 | 197 | −3 | 533 | 5 | 1,442 | 13 | 3,899 |
| −10 | 223 | −2 | 604 | 6 | 1,633 | 14 | 4,415 |
| −9 | 253 | −1 | 684 | 7 | 1,849 | 15 | 5,000 |
| −8 | 286 | 0 | 774 | 8 | 2,094 | | |

Audio was delivered through a Shure SRH240A closed stereo headphones. Sonifications of feedback about the position of a point and references that mark how far it is from an origin are discussed below.

### Pitch-only mapping

In the first design, we sonified the position of a point on an axis by mapping the pitch of a sine tone to the point's $Y$ coordinate following a positive polarity. That is, the tone's pitch changes in accordance with the point's movements on the axis; moving the point up increases the pitch, moving it down decreases it. We used an exponential function to map the position of the point to frequencies in the range of 120 Hz (for position −15) to 5,000 Hz (for position 15). The range and mapping were chosen to fit within the human hearing range, with the exponential distribution, subsequent frequencies differ by a constant factor instead of a constant term and this has been found to be superior to linear mappings (*Meijer, 1992*). Table 1 shows the pitch values for each point on this scale. Interaction with this sonification was designed such that the point moves when users press and hold a cursor key and not in response to single keystrokes. Pressing and holding a cursor key would therefore trigger a continuous progression of the sonified points being traversed as the point moves up or down on the axis.

### One-reference mapping

In the second design, we used the same pitch mapping described above and added one tone to convey a reference to an origin point. In this case, the reference tone represented the middle point on the scale (position 0 at a pitch frequency of 774 Hz lasting 100 ms). We designed this such that the user hears pitch changes that correspond to the movement of the point when they press and hold a cursor key, and hears the reference tone with a static pitch on key release. Comparing the two pitches (on key pressed and on key released) is meant to provide a sense of distance between the current position on the axis and the origin point based on pitch difference; the larger the difference in pitch between the two points the further away from the origin the point is located.

### Multiple-references mapping

In the third design, we again used the same pitch mapping as described above. But, instead of hearing only one reference point on key release, the user hears multiple successive reference tones with varying pitches that correspond to all the points between the current

position and the origin reference. Previous research has shown that the threshold for determining the order of temporally presented tones is from 20 to 100 ms (*Fraisse, 1978*). To create a succession of tones, our reference tones lasted 50 ms and were interleaved by a delay also of 50 ms. In this case, the position of a point in relation to an origin can be estimated by judging both the pitch difference at that point compared to the subsequent points, and the length of the sum of successive tones that separate it from the origin. A longer distance yields a longer succession of tones. Points located below the origin trigger an ascending set of tones, while those above the origin trigger a descending set of tones. For example, on reaching position 7, users hear a descending succession of tones made up of all the pitches of points 6, 5, 4, 3, 2, 1 and 0, the origin.

## Experimental design

We manipulated sonification and display type as independent variables in a between-subjects experimental design. Participants were divided into three groups; each group performed a set of point estimation tasks using one of the three sonification designs (between-subjects). To provide baseline comparisons, all participants performed two further sets of point estimation tasks under two control conditions (within-subjects): because our participants were sighted we chose to include a visual control condition; because speech provides more accurate position information compared to non-speech output, we chose to include a speech-only control condition. No sound was displayed in the visual control condition and participants could see the points as they moved them on the axis. In the speech-only control condition, participants received spoken feedback about the position value of a point. We used the Windows Text-to-Speech Engine (TTS) to speak the position values at the rate of 0.3 Words Per Second (WPS). The spoken numbers were also organised on a scale of 30 (from $-15$ to 15, the value 0 being the middle position). The order of conditions for each participant was balanced using a Latin Square design to compensate for any effects within trials. Each participant performed 22 trials per condition, totalling 66 trials per participant; thus giving 1,320 trial per condition and a total of 3,960 points for the whole experiment.

### Point estimation task

The task to be completed in each trial was to move a point on the $y$-axis to a target position and to do so as accurately as possible. The task involved:

- Looking at a target position (in all conditions)
- Estimating its position based on its visual position on the axis (Fig. 1)
- Using the keyboard arrow key to move the test point to the estimated position (by relaying on the visual, speech or sonification display).

In each trial, participants were first presented with a visual representation of the target position and were asked to memorise it. When the participants indicate they are ready to proceed, the system generates a random starting point on the $y$-axis from which the participants are required to use the cursor keys to move to the previously memorised target position (see Fig. 1). Participants pressed on a "next" button to move to the next trial. In the non-visual conditions (Speech, Pitch, OneRef and MultRefs), participants could see

**Table 2 Procedure.** Experimental procedure for one participant.

| Step | Description |
|------|-------------|
| 1 | Introduction to the experiment |
| 2 | Sign consent form and complete initial questionnaire |
| 3 | Training on the interface used in the first condition |
| 4 | Complete 22 Trials in the first condition |
| 5 | Training on the interface used in the second condition |
| 6 | Complete 22 Trials in the second condition |
| 7 | Training on the interface used in the third condition |
| 8 | Complete 22 Trials in the third condition |
| 9 | Informal interview |

[1] This study was approved by Queen Mary Research Ethics Committee. Approval number QMREC1356a.

the initial target position (without the reference tone(s)) but were presented with a blank screen hiding the randomly generated starting point and the axis when performing the second part of the task (i.e., they were unable to see the visual display). In this case, they had to rely on the spoken display or the sonifications to estimate the position of the point as they moved it to the target position. No accuracy feedback was provided between trials. Points positions were randomly generated by a computer program designed to ensure comprehensive coverage of points distribution along the axis across the 22 trials.

### Participants

We recruited 60 sighted participants to take part in this experiment (29 men and 31 women). The mean age was 26 (SD = 6.49). Participants were recruited through various means, including mailing lists and leaflets distributions. They were a mixture of university staff (both academic and non-academic), undergraduate and postgraduate students, and members of the public. All participants received a cash incentive for their participation.

### Procedure

Table 2 shows the steps of the experimental procedure for each participant.[1] Upon arrival, participants were provided with an overview of the experiment and were asked to complete an initial questionnaire that asked them about demographic details, their musical training (in terms of years of practice), their experience with non-visual interaction, and they were tested to establish whether or not they had perfect pitch perception. A total of 25 participants rated their musical training as beginner, 17 as intermediate, 6 experts and 12 had no prior musical training. Participants had no prior experience with non-visual interaction, and only one participant had perfect pitch perception. Participants were then randomly assigned to one of the three groups with the exception that care was taken to ensure that the different musical abilities were broadly equally distributed between the groups. Participants were then asked to complete 22 trials per condition (visual, speech, and one of the non-speech sonification conditions). Before the trials began, participants were trained on the particular display they were going to use and were allowed to spend as much time as they wished to get familiar with the interfaces. In particular, participants were introduced to the different sonification mappings used and instructed to spend as much time as they needed until they felt familiar with the mappings used. Once

familiar with the interfaces, participants then performed 4 trials similar to the actual trials used in the testing phases. Training typically lasted from 2 to 10 minutes per condition. We conducted informal interviews with the participants at the end of all the trials in order to discuss their impressions, preferences and experience. An entire session lasted between 30 min to 1 h per participant.

### *Dependent variables*

The dependent variables were point estimation errors and target selection time. Point estimation errors were measured as the difference between estimated points' positions and the target positions. Target selection time was measured as the duration from the first keystroke pressed when moving a point to the instance the "next" button press was logged.

### Hypotheses

The main hypotheses of the experiment were:

> H1: Participants will make significantly more point estimation errors when using the pitch-only sonification mapping compared to the one-reference and the multiple-references mappings.
>
> H2: Participants will make significantly more point estimation errors when using the one-reference sonification mapping compared to the multiple-references mapping.
>
> H3: Participants will be significantly slower at point estimation tasks when using the multiple-references sonification mapping compared to the pitch-only and the one-reference mappings.
>
> H4: The one-reference and multiple-references mappings will yield better performances for estimating points near the origin.

## RESULTS

We used single-factor repeated measures ANOVAs with display type as a factor (3 levels: visual, speech, and sonification) and a confidence level of $\alpha = 0.05$ to analyse data within groups against control conditions. We used Student $t$-tests when a statistically significant effect was detected to reveal differences between pairs. To analyse data across the three sonification conditions, we used single-factor independent measures ANOVAs with sonification type as a factor (three levels: pitch-only, one-reference, and multiple-references) and a confidence level of $\alpha = 0.05$. We used Tukey tests (HSD, 95% confidence level) and Bonferroni corrections when a statistically significant difference was found to reveal differences between sonification conditions.

### Point estimation errors within groups

Figure 2 shows the mean point estimation error for each sonification condition as compared to the visual and speech control conditions in each group.

### *Group 1: pitch-only mapping*

The ANOVA test for point estimation errors for Group 1 showed a significant main effect for display type ($F(2,38) = 66.589, p < 0.001, \eta^2 = 0748$). Pairwise Student $t$-tests showed that participants made significantly less errors when using the visual display

## Mean Point Estimation Error

**Figure 2** **The mean point estimation error within groups.** Error bars represent the Standard Deviation (Group 1: Pitch-only, Group 2: One-Ref, Group 3: Multi-Refs).

$(M = 0.67, SD = 0.07)$ compared to the pitch-only mapping condition $(M = 2.77, SD = 0.2)$ $(t = -2.566, p = 0.019)$, and when using a speech display $(M = 0.96, SD = 0.1)$ compared to the pitch-only mapping condition $(t = -8.547, p < 0.001)$. Differences between the visual and the speech control conditions were also statistically significant $(t = -8.626, p < 0.001)$.

### Group 2: one-reference mapping
Similarly, the ANOVA test for point estimation errors for Group 2 showed a significant main effect for display type $(F(2, 38) = 45.901, p < 0.001, \eta^2 = 0.705)$. Pairwise Student $t$-tests showed that participants made significantly less errors when using the visual display $(M = 0.52, SD = 0.16)$ compared to the one-reference mapping condition $(M = 1.96, SD = 0.93)$ $(t = -2.806, p = 0.011)$, and when using the speech display $(M = 0.67, SD = 0.22)$ compared to the one-reference mapping condition $(t = -6.784, p < 0.001)$. Differences between the visual and the speech control conditions were also statistically significant $(t = -6.947, p < 0.001)$.

### Group 3: multiple-reference mapping
The ANOVA test for point estimation errors for Group 3 also showed a significant main effect for display type $(F(2, 38) = 7.425, p < 0.002, \eta^2 = 0.586)$. Pairwise Student $t$-tests showed that participants made significantly less errors when using the visual display $(M = 0.85, SD = 1.15)$ compared to the multiple-references mapping condition $(M = 1.77, SD = 1.03)$ $(t = -2.518, p = 0.021)$, and when using the speech display $(M = 0.74, SD = 0.33)$ compared to the multiple-reference mapping condition $(t = -4.508, p < 0.001)$. Differences between the visual and the speech control conditions were not statistically significant in this case $(t = -0.401, p < 0.693)$.

**Figure 3** **The mean target selection time (milliseconds) within groups.** Error bars represent the Standard Deviation (Group 1: Pitch-only, Group 2: One-Ref, Group 3: Multi-Refs).

## Point estimation errors across groups

The ANOVA test for point estimation errors across groups showed a significant main effect for sonification type ($F(2, 57) = 5.908, p = 0.005, \eta^2 = 0.127$). Post hoc pairwise comparisons using Tukey HSD showed that participants who used the pitch-only mapping made significantly more point estimation errors ($M = 2.77, SD = 0.95$) when compared to participants who used the one-reference mapping ($M = 1.96, SD = 0.93$) ($p = 0.03$) and when compared to participants who used the multiple-references mapping ($M = 1.77, SD = 1.33$) ($p = 0.006$). There was no significant difference between the one-reference and the multiple-references mappings ($p = 0.806$) (see Fig. 2). These results support hypothesis H1 and reject hypothesis H2.

## Target selection time within groups

The ANOVA tests showed that, for all three groups, there was a significant main effect of display type on target selection time (Group 1 $F(2, 38) = 33.224, p < 0.001, \eta^2 = 0.674$, Group 2 $F(2, 38) = 73.601, p < 0.001, \eta^2 = 0.911$, Group 3 $F(2, 38) = 59.936, p < 0.001, \eta^2 = 0.732$). Pairwise Student $t$-tests showed that participants in all three groups were significantly faster at estimating the position of points on the visual control condition when compared to both the speech condition and the corresponding sonification condition. Participants in all three groups were also significantly faster in the speech control conditions when compared to the sonification conditions (see Fig. 3).

## Target selection time across groups

The ANOVA test for target selection time across groups showed a significant main effect for sonification type ($F(2, 57) = 6.577, p = 0.003, \eta^2 = 0.233$). Post hoc pairwise comparisons using Tukey HSD showed no significant effect between participants who used

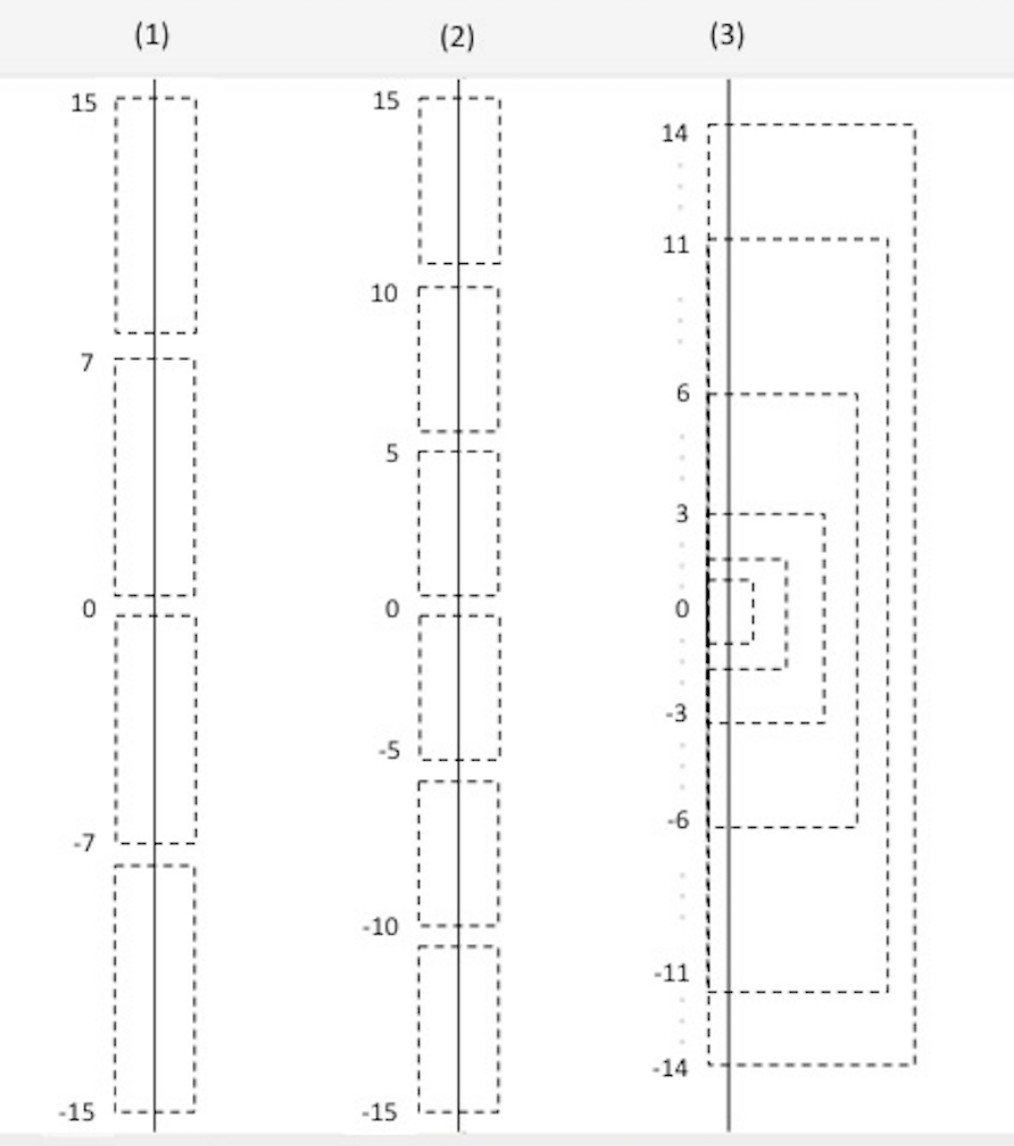

**Figure 4** **Range-based analysis.** The various groupings we explored to analyse point estimation errors within target ranges.

the pith-only mapping and the one-reference mapping ($p = 0.975$). However, there was a significant difference between target selection times between participants who used the multiple-references mapping and the pitch-only mapping ($p = 0.006$) and between the multiple-references mapping and the one-reference mapping ($p = 0.01$). As shown in Fig. 3, participants who used the multiple-references mapping were significantly slower than those who used the other two sonification mappings. These results support hypothesis H3.

## Point estimation errors within ranges of target positions

To test hypothesis H4, we examined differences in point estimation errors across groups within a variety of target ranges. We explored a number of strategies for dividing the scale

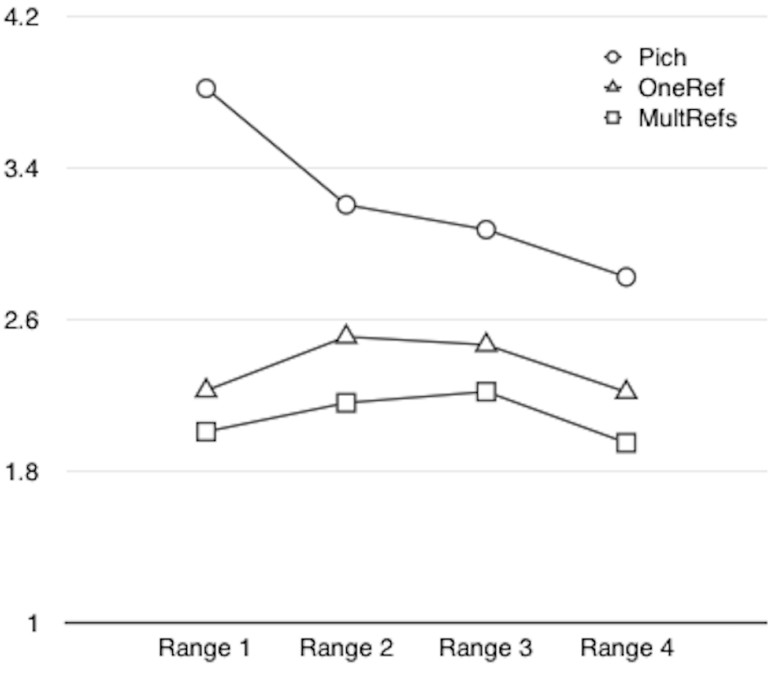

**Figure 5** Effects of interactions between range and type of sonification.

and grouping target positions as shown in Fig. 4. In the first strategy, we divided the scale into four equal segments at positions 0, 7 and −7. In a second strategy, we divided the scale into six equal segments at positions 0, 5, −5, 10 and −10. In the third strategy, we gradually increased the range of target positions considered in the analysis starting from −1 and 1 all the way to −14 and 14. We ran a two-way mixed ANOVA to examine the effects of interactions between range (within-subjects variable) and type of sonification (between-subjects variable) on participants performance on point estimation tasks. The aim was to examine whether and where on the scale range would a given type of sonification be most effective.

Figure 5 summarises the results we obtained from analysing point estimation errors using the third grouping strategy, which was particularly effective at revealing differences between the sonification conditions. In particular, analyses showed distinct differences in performances across four ranges of target positions; Range 1 encompassing target points between positions −3 and 3, Range 2 for target points between −5 and 5; Range 3 for target positions between −11 and 11; and Range 4 for target points between −14 and 14. Target points located within the range of −1 and 1 were excluded from the analyses because this range did not contain enough data points to reliably run the statistical tests. There was no significant main effect of range ($F(3, 171) = 1.972, p = 0.159$) and no significant range × type of sonification interaction ($F(6, 171) = 1.963, p = 0.131$). There was a significant main effect of sonification type ($F(2, 57) = 3.23, p = 0.047, \eta^2 = 0.102$). Results of simple effects were as follows:

- Range:
  1. Pitch sonification: There was a significant main effect of range ($F(3, 55) = 4.339, p = 0.008, \eta^2 = 0.191$), in particular participants made significantly more errors in range 1 compared to ranges 2, 3 and 4 ($p = 0.001, p = 0.029, p = 0.009$, respectively). They also made significantly more errors in Range 3 compared to Range 4 ($p = 0.017$).
  2. One-Reference sonification: There was a significant main effect of range ($F(3, 55) = 3.766, p = 0.016, \eta^2 = 0.17$), with participants making significantly more errors in Range 3 compared to Range 4 ($p = 0.018$).
  3. Multiple-References: There was a significant main effect of range ($F(3, 55) = 3.223, p = 0.029, \eta^2 = 0.15$), also with participants making significantly more errors in Range 3 compared to 4 ($p = 0.01$).
- Sonification type:
  1. Range 1: There was a significant main effect of sonification type ($F(2, 57) = 3.96, p = 0.025, \eta^2 = 0.122$). In this range, participants who used the pitch-only sonification made significant more errors than those who used the one-reference ($p = 0.027$) and the multiple-references sonifications ($p = 0.013$).
  2. Range 2: There was no significant main effect of sonification type on participants performances in Range 2 ($F(2, 57) = 1.156, p = 0.218$).
  3. Range 3: There was no significant main effect of sonification type on participants performances in Range 3 ($F(2, 57) =, p = 0.093$), however participants using the pitch-only sonification made significantly more errors than those who used the multiple-references sonification ($p = 0.035$).
  4. Range 4: There was a significant main effect of sonification type on participants performances in this range ($F(2, 57) = 3.99, p = 0.024, \eta^2 = 0.052$). Participants who used the pitch-only sonification made significantly more errors than those who used the one-reference ($p = 0.06$) and the multiple-references ($p = 0.008$) sonifications.

Overall, the above results provide partial support for hypotheses H2 and H4. They show that participants' performances using the multiple-references mapping was fairly consistent across the scale, outperforming the pitch-only mapping in all but one range of target points (Range 2). On the other hand, performances using the one-reference were less consistent, failing to outperform the pitch-only mapping within ranges 2 and 3 of target points.

## DISCUSSION

The goal of the experiment presented in this paper was to contrast different ways of conveying reference markers using sonification and examine the effects that these have on users when performing non-visual point estimation tasks. The hypotheses of the experiment addressed the question of what effect does the addition of reference tones have on users' performance when using different types of sonification-based displays. With regards to performances in the control conditions, the results from the experiment showed that participants performance on point estimation tasks was significantly affected when using the sonification displays. Point estimation errors were lowest in the visual condition, which was anticipated given that our participants were sighted and had no prior experience

with non-visual interaction. Errors were also significantly lower when participants used a speech-based display compared to sonification conditions, which we also anticipated because of the precise nature of spoken feedback.

In relation to the main question addressed, the results showed that there were differences between performances across the three sonification conditions, suggesting that the way reference markers are conveyed using these sonifications does affect target accuracy. Users made significantly more point estimation errors when using the pitch-only sonification mapping compared to the one-reference and the multiple-references mappings. Meanwhile, the mean error across the whole range of the scale investigated in this experiment was similar for one-reference and multiple-references mappings. This shows that the addition of reference markers is important and needs to be taken into consideration when designing non-visual interaction techniques. Similar findings were reported for adding context information when sonifying graphs (*Smith & Walker, 2002*; *Nees & Walker, 2008*) and scroll bars (*Yalla & Walker, 2008*). However, previous research has only explored the addition of context in the case of passive listening. The experiment presented in this paper shows that such benefits extend to interactive sonification where users have active control over the audio output as they explore a data set from an unknown starting point.

There were also differences between the three sonification mappings when compared in terms of target selection times. The results showed that participants were significantly slower at estimating the position of points when using a multiple-references mapping. We had anticipated that participants will be faster when using the pitch-only and the one-reference mappings. This was because reference markers in the multiple-references mapping are presented by aggregating a succession of tones of 50 ms each, which automatically results in a lengthier display compared to a single 100 ms tone in the one-reference mapping. However, the informal interview discussions also revealed that participants tended to spend more time interpreting the sonified reference information they received through the multiple-references mapping. Combined with the proportion of point estimation errors highlighted above, these results explain participants' superior performances under this condition and suggest that there is a trade-off between speed and accuracy related to the amount of reference information to embed in a sonification of this kind. Designers should therefore take this trade-off into consideration when designing sonification of reference information.

For a more in-depth analysis, we examined how point estimation errors differed across various ranges of target positions. Participants who used the pitch-only mapping made significantly more point estimation errors across all target positions with the exception of the range between −5 and 5 on the scale. Analysis of performances using the one-reference and multiple-references mapping revealed more varied results. The mean error between these two sonification mappings was similar for target positions near the origin and near the extreme ends of the scale. However, the mean error between the one-reference and the pitch-only mappings were similar for target positions within the middle ranges of the scale (from −6 and 6 to −11 and 11). The performance of participants who used the one-reference mapping was therefore not as consistent as those who used the multiple-references, which consistently outperformed the pitch-only mapping across these ranges.

We referred back to the subjective data gathered from the informal interviews to further explore the reasons that might explain the above results. Participants seem to have found it difficult to estimate target positions in the middle ranges of the scale due to the lack of accurate reference information. When close to the origin, it was easy to either count the number of tones in a succession of multiple reference tones or to judge pitch differences between the tone of a point's position and the origin tone. However, as the succession of tones and pitch differences increased, counting tones and comparing pitch differences became more difficult and less accurate. Multiple references contained too many points that were presented too quickly to be counted, and pitch differences were too far apart to be judged accurately. Thus, the analysis of participants' point estimation errors in these middle ranges seems to suggest that not only was there a threshold at which the information conveyed through sonified reference markers became less accurate—and hence less useful—but also that such a threshold was different for the multiple-references and the one-reference mappings. Multiple reference tones continued to give useful information throughout the middle ranges, while the one reference tone became less accurate as soon as the target position moved farther from the origin. This is also illustrated by the similarity in mean errors between the pitch-only and the one-reference mapping in target positions located in the middle of the scale.

Interestingly, participants also commented that they did not often rely on the sonified reference markers when targeting points near the extreme ends of the scale (from −12 and 12 to −14 and 14). This confirms that multiple reference tones, while useful across the middle ranges, still reached a threshold where the information they conveyed became redundant. These findings confirm those reported elsewhere in the literature (*Smith & Walker, 2005*). Although they do not explain why participants who used the multiple-references mapping still outperformed those who used a pitch-only mapping in this range of target positions, they suggest a more dynamic redesign of the sonification mappings. For example, while adding reference information is important, being able to switch the point of origin from the middle of the scale (representing 0 in our case) to mark other areas of interest, such as the extreme values, might improve point estimation. This technique would be similar to the use of auditory beacons for audio-only navigation (*Walker & Lindsay, 2006*), and our results show that such a technique might be successfully adapted to support orientation within sonification of graphs. This technique might only be applicable when target positions are known in advance, however, which means that providing reference to one or more static origins is still important for exploratory interaction.

Overall, the above findings contribute to research on non-visual interaction with graphs by extending relevant research with the investigation of active rather than passive point estimation tasks (e.g., *Smith & Walker, 2002*; *Nees & Walker, 2008*), i.e., where users have direct control over the auditory display as they estimate the position of a given point. Also, existing work that investigated audio-haptic interaction with graphs did not explicitly address the question of support for point estimation tasks (e.g., *McGookin & Brewster, 2007*). Given the nature of the task examined in the presented study, our findings can be used to support better designs of graphing and drawing applications, where point estimation forms an integral part of editing graphs and sketching, as well as more general applications

involving pointing and target acquisition actions, such as interfaces for virtual and augmented reality (*Li, Dearman & Truong, 2010*; *Fiannaca, Morelli & Folmer, 2013*) and for audio-haptic and tangible interactions with graphed data. There are limitations to our findings, however. Results may differ for scales larger than the one used in this experiment. In particular, it is likely that larger scales would require more references and context information embedded in the axis to ease navigation and orientation. Additionally, we have restricted our investigation to one-dimensional vertical movements. Results may differ if sonification of reference markers are used to support two-dimensional movements. Finally, it would also be interesting to examine different types of sonifications, e.g., using a musical scale, and how these compare to the exponential sonification used in the presented experiment, and to investigate the impact of physical muscle memory over the trials as a possible confounding variable.

## CONCLUSIONS

We presented an experiment that examined the effects of adding reference tones when designing sonifications to support non-visual point estimation tasks. Our results showed that adding context information in the form of reference markers is important for supporting navigation and exploration, and that care must be taken to account for thresholds of information redundancy. In our case, multiple references as a succession of tones was useful for providing a sense of distance from a target position but became redundant when conveying long distances. Similarly, a single static pitch reference was also useful, but harder to interpret for targets farther away from a static origin. We also found that using multiple reference tones supported more consistent superior performance on point estimation tasks particularly in the middle ranges of an axis, and recommended that sonification of reference information should be designed to account for a speed/accuracy trade-off and allow for dynamic control of reference direction to account for both known and unknown target points. These findings have implications for the design of auditory graphs and more generally for interfaces that require target acquisition and employ interactive sonification as a display technique.

### Funding
This work was funded by the EPSRC Grant number EP/J017205/1. The funders had no role in study design, data collection and analysis, decision to publish, or preparation of the manuscript.

### Grant Disclosures
The following grant information was disclosed by the authors:
EPSRC: EP/J017205/1.

### Competing Interests
The authors declare there are no competing interests.

## Author Contributions

- Oussama Metatla conceived and designed the experiments, performed the experiments, analyzed the data, wrote the paper, prepared figures and/or tables.
- Nick Bryan-Kinns and Tony Stockman conceived and designed the experiments, reviewed drafts of the paper.
- Fiore Martin conceived and designed the experiments, performed the computation work.

## Ethics

The following information was supplied relating to ethical approvals (i.e., approving body and any reference numbers):

Queen Mary Research Ethics Committee; Approval number QMREC1356a.

## Data Availability

The raw data has been supplied as Data S1.

## Supplemental Information

Supplemental information for this article can be found online at http://dx.doi.org/10.7717/peerj-cs.51#supplemental-information.

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
