# Peer review of "Sonification of reference markers for auditory graphs: effects on non-visual point estimation tasks"

_PeerJ Computer Science, doi:10.7717/peerj-cs.51_

## Round 0.1 · original submission · Major Revisions

Please address the issues raised by the reviewers. In particular please conduct a new statistical analysis of the data that allows you to explore the interactions between variables and decreasing the risk of over-testing. This may well change your results and discussion sections.

·

Basic reporting

The manuscript was generally well-researched and well-written. The topic is interesting and relevant to the auditory display research community, and the execution of the research was competent. Overall, this manuscript was clear, concise, and enjoyable to read, and I think it offers a worthwhile contribution to the literature.

Experimental design

I have one major concern that, if addressed, could potentially strengthen the precision/clarity of the manuscript. I was not totally clear on the task structure (section 2.2.1). I think more detail should be provided. Was the target tone presented visually, with sound, or both? Or did the presentation of the target tone match the presentation format of the test condition (speech for speech, etc.)? Was a target tone for the reference conditions presented with references or as a single tone without references? As participants responded, did they always see the visual display? If yes, could they have visually anchored the target and simply used the visual slider to match its remembered position in space? Further, did the method of interaction with the keyboard possibly influence results in addition to the auditory display, since the interaction method was (necessarily) slightly different across conditions? The task is described as a point estimation task, but, if I understand correctly, the task required: 1) remembering the target tone; and 2) using the keyboard to move the test tone until it matched the remembered target tone. As such, the current description in 2.2.1 makes the task sound more like a perceptual matching task—sort of like a method of adjustment approach--than a point estimation task. Point estimation usually requires the participant to generate and assign a value to the magnitude/quantity represented by the sonification. I don’t see this as a critical flaw, but I think a clarification of the task (and if necessary a reconsideration of whether it actually involved point estimation or instead was more of a memory matching task) could strengthen the clarity and precision of the manuscript. As a related point, I think the Discussion would be strengthened if the manuscript linked the study task to specific potential applications of this type of task.

Minor comments:

The design was described as mixed, but it was more of a series of within subjects designs, with one level of each of those separate designs compared between groups. This was a somewhat unusual approach, because it required all of the work of a factorial design without the benefit of having the extra data for all 9 conditions. The separate speech and visual conditions across groups shown in Figure 1 have no substantive difference between them (if I understand correctly), thus differences in these conditions across groups probably represent unreliability in measurement to the extent that they are different (though they look fairly similar, so that’s good). Because each sonification condition included two control conditions, the analysis also was forced to proceed with multiple individual ANOVAs for each group. A more economical approach, both experimentally and statistically, would have been a 1X5 design (each of the 3 sonification types plus the 2 control conditions), probably in a between subjects design. This comment is more of a suggestion for future research. I don’t think it substantively affected the interpretation of any of the findings.

Effect sizes would be nice to report for significant results.

In 2.2.4, the DV was the absolute value of the difference, correct? This should be clarified. Correspondingly, the Y axis in Figure 2 shows negative values that should be removed from the figure if no negative values were possible.

Yalla and Walker used a design that was similar to the manuscript’s one-reference mapping. They called it a “double-tone” auditory scroll bar. It might be worth consulting:
Yalla, P., & Walker, B. N. (2008). Advanced auditory menus: Design and evaluation of auditory scroll bars. Proceedings of the Tenth International ACM SIGACCESS Conference on Computers and Accessibility (ASSETS2008), Halifax, Canada (13-15 October, 2008). pp. 105-112.

I wanted more explanation/extrapolation after the single sentence that was at the end of 4.1.4.

Validity of the findings

I thought the corroboration and discussion of the results in the context of the debriefing interviews was helpful. I appreciate that the authors got the most explanatory value from all data they collected. I also thought the exploratory detailed analyses of response patterns across conditions and positions on the response scale was novel and insightful.

Additional comments

The manuscript was quite well-written and clear overall. There were a couple of typos that I noticed, however (e.g., “focusing” misspelled in 2.1, sentence fragments beginning with “for example” in paragraph 1 and “for instance” in 1.1, paragraph 2), so the authors will want to give a final proof before the version of record is published.

·

Basic reporting

There is an additional "Further Discussion" section which appears to deviate from the standard form for Peerj.

There are small typos throughout (at one point we have a '?' in an example).

There seems to be segments of the literature that are missing, in particular work from the augmented reality sphere (which is relevant given the device being used in the experiment). Examples include Air Pointing by Cockburn, Gutwin et al., haptic target acquisition by Fiannca. These possibly are relevant.

Experimental design

The introduction of speech as a control condition seems to come out of no where. It is not clear why it was introduced.

Validity of the findings

The data is robust, however the analysis is problematic in a couple of different ways:

1) By separating your variables out into one way ANOVAs you miss the potential for (interesting) interaction effects between the different variables.
2) The introduction of the ranges as a variable is interesting; however, again you are separating each out where there may be interactions that are happening with the other variables. As a result you are also over testing - each test has the potential of finding an effect and you are compounding that with each of the ranges. You should be able to integrate these ranges as emergent variables into your analysis.

There is a lack of discussion about limitations in the design, the possible confounds that may impact the results (such as physical muscle memory over the trials).

---

## Round 0.2 · accepted · Accept

Thank you for your careful consideration of the comments from the reviewers and your thorough revision of the paper.

·

Basic reporting

I noticed one typographical error that should be corrected in the final version: on page 12, line 5, "However, previous research have only..." should be "research has."

Experimental design

To the extent possible, the authors have addressed all of my comments in the revision.

Validity of the findings

To the extent possible, the authors have addressed all of my comments in the revision.

Additional comments

I appreciate the authors' thoughtful responses to the comments in my original review.